# The Promising Role of Antioxidant Phytochemicals in the Prevention and Treatment of Periodontal Disease via the Inhibition of Oxidative Stress Pathways: Updated Insights

**DOI:** 10.3390/antiox9121211

**Published:** 2020-12-01

**Authors:** Thi Thuy Tien Vo, Pei-Ming Chu, Vo Phuoc Tuan, Joyce Si-Liang Te, I-Ta Lee

**Affiliations:** 1School of Dentistry, College of Oral Medicine, Taipei Medical University, Taipei 110, Taiwan; vtthuytienrhm@gmail.com; 2School of Medicine, College of Medicine, China Medical University, Taichung 406, Taiwan; pmchu@mail.cmu.edu.tw; 3Endoscopy Department, Cho Ray Hospital, Ho Chi Minh City 700000, Vietnam; vophuoctuandr@gmail.com; 4Department of Medical Education, Kaohsiung Chang Gung Memorial Hospital, Kaohsiung 833, Taiwan; u1020001511@cmu.edu.tw

**Keywords:** oxidative stress, antioxidant, periodontal disease, oral cavity, inflammation, oxidants

## Abstract

There is growing evidence on the involvement of oxidative stress, which is simply described as the imbalance between oxidants and antioxidants in favor of the former, in the development of periodontal disease that is the most common inflammatory disease in the oral cavity. Thus, the potential of antioxidant phytochemicals as adjunctively preventive and therapeutic agents against the initiation and progression of periodontal disease is a topic of great interest. The current review firstly aims to provide updated insights about the immuno-inflammatory pathway regulated by oxidative stress in periodontal pathology. Then, this work further presents the systemic knowledge of antioxidant phytochemicals, particularly the pharmacological activities, which can be utilized in the prevention and treatment of periodontal disease. Additionally, the challenges and future prospects regarding such a scope are figured out.

## 1. Introduction

Periodontal disease comprises a group of all conditions, from inflammation to infections to tumors, that affect the well-being of the tooth-supportive apparatus, the so-called periodontium, including the gum (gingiva), periodontal ligament, cementum, and alveolar bone [1]. Among the wide variety of periodontal illnesses, gingivitis and periodontitis are the most common and have predominant relevance to both oral health and overall health. If left untreated, these diseases are highly likely to cause severe consequences for oral structures, such as bone resorption, tooth mobility, and eventually tooth loss, thereby compromising the functional integrity as well as appearance and social interaction. More seriously, the adverse outcomes of periodontal disease are not only restricted within the oral cavity but also even go beyond. It can negatively affect other organ systems and cause unexpected events in medically compromised patients [2]. Accumulating evidence regarding the relationship between periodontal disease and various systemic conditions, such as cardiovascular diseases, diabetes, arthritis, Alzheimer’s, pregnancy complications, and a growing list of other conditions, has been reported in the literature [3,4]. Despite such deleterious effects, periodontal disease has still been overlooked in many countries and regions, particularly in those with low socioeconomic status and a weak healthcare system, making it widespread in communities. It was reported that up to 90% of the global population have experienced some forms of periodontal disease in their life [5], and up to 50% have been affected by these illnesses [6]. According to the latest data from the Global Health Data Exchange database, periodontal disease is currently the 12th most prevalent condition worldwide. Furthermore, the cumulative prevalence has increased by 34% during the past three decades, and it is expected to increase in coming years [7]. Remarkably, periodontal disease is both preventable and treatable through adequate and proper approaches. One of the key fundamentals for such modalities is the management of causal factors. It is well established that the major etiologic factor of periodontal disease is periodontally pathogenic bacteria colonizing the dental plaque or biofilm, particularly concerning the subgingival area. Therefore, the control and eradication of such microorganisms as well as microbial biofilm, e.g., improving oral hygiene routine combined with periodically having professional dental cleaning, are considered as the most effective regimens for prevention and treatment of these illnesses [5]. Besides, the immuno-inflammatory responses involved in the onset and progression of periodontal disease have also been paid attention in recent years. It is well known that inflammation is one of the physiological events in response to offenses, such as stress, heat, chemical agents, infection, and injuries. Thus, the initial inflammation in the periodontal tissues can be considered as the physiological defense combating the microbial challenge and initiating the healing processes rather than the pathological response. However, if it is prolonged and severe, this inflammation might lead to tissue breakdown. Otherwise speaking, periodontal disease also arises from the dysregulation or dysfunction of the activation and resolution pathways of inflammation, subsequently resulting in chronic and destructive inflammation [8]. The underlying mechanisms are yet to be elucidated. A growing body of evidence has suggested that oxidative stress appears to be the culprit of both local (periodontal in this case) and systemic inflammation [9,10,11]. Oxidative stress represents the increase of oxidants and/or the decrease of antioxidants, leading to the damage of vital cellular components and the destruction of tissues via different paradigms, such as lipid peroxidation and protein inactivation [11]. Hence, the oxidative stress-mediated inflammation is likely to be one of the plausible pathways contributing to the development of periodontal disease. This implies that the conventional prevention and treatment of periodontal disease focusing on the management of bacterial pathogens seem to be insufficient, and the oxidative stress-reducing regimens using antioxidative supplementation have emerged as promising preventive and therapeutic adjuncts for these illnesses. It is noteworthy that up to 80% of the global population use herbal products, such as extracts, teas, and other active compounds, for their basic health care [12]. In fact, herbal medicine has been utilized in every country to prevent and treat diseases throughout history. A growing body of evidence indicates herbs as sources of a variety of phytochemicals, many of which possess powerful antioxidant activities [13]. Moreover, herbal medicines are preferred over chemical drugs due to their wider biological activities and higher safety but lower costs. As a result, natural products from herbal remedies may play a pivotal role in antioxidant defense and redox balance, suggesting their potency in the prevention and treatment of periodontal disease via the inhibition of oxidative stress pathways. In order to assist oral healthcare professionals in designing and implementing effective strategies that promote better periodontal health, this work begins with a brief introduction of the immuno-inflammatory pathogenesis of periodontal disease, and then presents a thorough discussion of oxidative stress along with its role in the immuno-inflammatory paradigm involved in periodontal disease. Finally, a series of updated findings on the potential role of phytochemicals with remarkable antioxidant properties that might be used for the management of periodontal disease is systemically presented.

## 2. Immuno-Inflammatory Pathogenesis of Periodontal Disease

In the context of this work, periodontal disease is defined as chronic inflammatory diseases, including gingivitis and periodontitis, that affect the tooth-supporting tissues. In general, gingivitis is described as a specific inflammation of gingiva (gum) mainly due to the accumulation of dental plaque or microbial biofilm. This condition is typically reversible upon the removal of the bacterial challenge, and causes no severe problems until it has become advanced. Unlike gingivitis, periodontitis is a destructive inflammation, which is characterized by the breakdown of periodontal tissues, such as periodontal ligament and alveolar bone, and it is irreversible [14]. The major etiology of periodontal disease is typically attributed to the invasion of microorganisms located within the oral microflora, particularly the subgingival dental plaque. In the past, gingivitis and periodontitis used to be considered as a continuum in which chronic gingivitis eventually progresses to destructive periodontitis over time. Nevertheless, in fact, this progression does not always occur, implicating that the microbial biofilm is essential to the etiology and pathogenesis of periodontal disease but is insufficient. Owing to the presence of periodontal microbes in individuals without evidence of disease development, the concern about the initiation and progression of these conditions has been dramatically focused on the net effect of the immune responses and the inflammatory processes in recent years [8]. This postulation is further supported by the fact that some diseases and systemic conditions or habits that modify the host response, such as diabetes mellitus, cardiovascular disease, obesity, medications, and tobacco smoking, among others, are risk factors or damage accelerators for periodontal disease [15]. Currently, the existing literature has indicated that both microbial biofilm and periodontal inflammation play pivotal roles in the onset and development of periodontal disease, and they mutually reinforce each other. Upon their invasion, the periodontally pathogenic microorganisms produce enzymes and other substances, which are able to cause the breakdown of periodontal tissues directly. Meanwhile, the cellular and humoral components of the host immune system, mainly associated to the periodontal immune responses, are activated. Typical examples of such protective aspects of the host response are the recruitment of neutrophils and macrophages, the production of cytokines and chemokines, and the activation of the complement cascade. If the immune defenses properly work, the periodontium is prevented from damages caused by the bacterial toxins and by-products. Thus, the periodontal inflammation can be considered as a local reaction against the microbial challenge in order to arrest or eliminate such causal factors. Otherwise speaking, the initial inflammation in the periodontal tissues may function as a physiological defense in response to the pathogens. Conversely, due to the incompetent immune or due to the persistent bacterial challenge, there is the failure of thoroughly eliminating the infection, and then the immune system is continuously stimulated, leading to chronic inflammatory responses. Such a perpetuation of inflammatory responses ultimately disrupts the homeostatic mechanisms and releases various mediators, such as proinflammatory cytokines, proteases, and prostanoids which can promote gingival breakdown and bone loss [8]. Therefore, it is increasingly obvious that the prospective strategies of periodontal healthcare will rely on a detailed map and perception at the molecular and cellular levels of the initiation and resolution of aberrant inflammation. To fill the gaps related to the immuno-inflammatory pathways in the pathogenesis of periodontal disease, there has been a growing body of publications providing panorama knowledge on the regulation of inflammatory responses in the periodontal tissues through the immunologic mechanisms [8,16,17]. Albeit the advances have been made in the immuno-inflammatory field of periodontal pathology, much more is still demanded for a complete understanding. Herein, our work addresses the role of oxidative stress, which can be simply described as the homeostatic imbalance between oxidants and antioxidants in favor of the former, in the model of immuno-inflammatory regarding periodontal disease.

## 3. The Role of Oxidative Stress in the Immuno-Inflammatory Paradigm of Periodontal Disease

### 3.1. Oxidants in Periodontal Tissues

Oxygen, or molecular oxygen (O_2_), is a key element for aerobic lives, including human. In eukaryotic cells, oxygen is completely reduced into water (H_2_O) through a four-step reduction given as the following: O_2_ → O_2_^•−^ → H_2_O_2_ → ^•^OH → H_2_O, via the electron transport chain (ETC) in mitochondria [18]. Such sequential reduction of O_2_ has arisen a matter of interest, which is referred to as the oxygen paradox. While the tetravalent reduction of O_2_ produces energy for aerobic organisms, its univalent reduction as a result of the leakage of electrons from the electronically reductive flow generates metabolites, the so-called reactive oxygen species (ROS), which can be divided into free radicals and non-radicals. A free radical is characterized by the presence of one or more unpaired electrons in its spatial configuration, making it highly unstable and reactive. Hence, free radicals can induce chain reactions through which other free radicals are consecutively generated. Free radicals of ROS that are of importance include superoxide anion radical (O_2_^•−^) and hydroxyl radical (^•^OH). The former is the most important widespread form as it leads to a cascade of other ROS production, whereas the latter represents the most reactive species. On the other hand, ROS also governs other non-radicals that are not true radicals but are capable of conversion into free radicals. Hydrogen peroxide (H_2_O_2_) is considered as the most significant non-radical of ROS since it converts into other ROS, particularly hydroxyl radical, in the presence of metals, such as iron, via Fenton reactions. Furthermore, hydrogen peroxide can generate hypochlorous acid (HOCl) in the presence of chloride anion, which is catalyzed by the unique enzyme in neutrophils namely myeloperoxidase (MPO). Hypochlorous acid is a strong reactive species, and it is grouped under the category of ROS as well [19]. Beyond the scope of the above species, nitric oxide (NO^•^ or NO) also represents a crucial pathway associated to redox intermediate production. It has been well documented that NO plays a significant role as a physiological mediator in various tissues and organs in the human body. Following the characterization of the messenger molecule within the endothelial cells with vasodilating potential, termed endothelial-derived relaxing factor (EDRF), the fact that NO can function as EDRF was demonstrated [20]. In addition to its biological actions in the vasculature, the role of NO in neurotransmission and the cellular immune system has been appreciated as well. The understanding about the biological functions of NO was comprehensively reported in a recent review [21]. Remarkably, NO plays dual roles in biological systems. In particular, NO at low optimal concentrations can regulate numerous physiological processes, whereas the overproduction of NO may contribute to the pathogenesis of many diseases [22]. NO is typically generated through the metabolism of the amino acid L-arginine catalyzed by nitric oxide synthase (NOS) enzymes. Up to date, three different isoforms of NOS have been identified, including neuronal NOS (nNOS or NOS I), inducible NOS (iNOS or NOS II), and endothelial NOS (eNOS or NOS III). It has been widely accepted that two isoforms, nNOS and eNOS, are constitutively expressed, whereas iNOS isoform is only induced in response to infection, inflammation, or trauma [23]. Interestingly, NO generated by NOS located in different cellular and subcellular locations may elicit specific effects [24]. The generation of NO by iNOS has been indicated to function as a microbial killer and immune regulator. Besides, inflammatory cells, such as neutrophils and macrophages, which can generate high levels of ROS, also play critical roles in host defense and immunity [25]. Thus, the interplay between NO and ROS in cellular redox signaling may regulate the immuno-inflammatory responses. From the viewpoint of redox biology, NO was initially underestimated despite its radical nature. Nevertheless, owing to the discovery that NO can rapidly react with superoxide to produce highly reactive peroxide, namely peroxynitrite (ONOO^−^), the merging of ROS and NO pathways has occurred, highlighting the importance of NO in the redox paradigm. Peroxynitrite itself is highly toxic due to its powerful oxidant capacity. Moreover, it can further undergo reactions to generate hydroxyl radical and other reactive nitrogen species (RNS) [26]. Conversely, there have been a number of studies reporting the shifting from NO production into ROS generation by iNOS, which is regulated by ROS [25,27]. The decrease of iNOS-induced NO generation by ROS is presumed to result from the ROS-caused oxidation of the pterin cofactor tetrahydrobiopterin (BH4), which is required for NO production by iNOS. In the absence of BH4, iNOS fails to generate NO and becomes uncoupled, whereby the reduction of molecular oxygen driven by electron flow from the NOS reductase domain can generate superoxide or other ROS [27]. However, the precise mechanisms concerning the interaction of NO and ROS are not fully understood, requiring further investigations in the future. Taken together, the term ROS covers all kinds of chemical species derived from molecular oxygen, including radicals, molecules, and ions, that act as major pro-oxidants or oxidants in the biological system. The major generator of ROS is the mitochondrial ETC by which premature electron leakage of oxygen reduction may occur, liberating superoxide. Superoxide then dismutates into hydrogen peroxide spontaneously or by the catalysis of superoxide dismutase (SOD). In turn, hydrogen peroxide is further transformed into other species by three paths: (1) reduced into water by catalase (CAT) or glutathione peroxidase (GPX), (2) degraded into hydroxyl radical via Fenton and Haber–Weiss reactions, and (3) converted into hypochlorous acid by MPO in neutrophils. Besides, enzymatic reactions catalyzed by enzymes, including nicotinamide adenine dinucleotide phosphate (NADPH) oxidases (NOX) and xanthine oxidases (XO), also play a significant role in ROS production. Other enzymes, such as cyclooxygenases (COX), lipoxygenases (LOX), and cytochrome P450 monooxygenase (P450s), have been proposed to generate ROS as well. However, none of these enzymes are typical ROS manufacturers, nor are superoxide and hydrogen peroxide primary reactive species produced during their catalytic activity [18]. Considering the oral cavity is one part of the whole body, the oral tissues, including periodontal tissues, are physiologically exposed to such sources of ROS. In addition to the physiological sources, ROS generation in periodontal tissues is enhanced upon the invasion of periodontal pathogens. As mentioned above, periodontal disease should be considered as the result of the interaction between pathogenic microorganisms and the immuno-inflammatory host responses. The lipopolysaccharides (LPSs) and other cell components from the microbes may lead to the production of various proinflammatory cytokines as well as the activation of signaling pathways, such as activator protein 1 (AP-1) and nuclear factor kappa B (NF-κB). Both routes are able to cause the recruitment and activation of immune cells, thus promoting ROS liberation. The former induces the recruitment and activation of hyper-responsive polymorphonuclear leukocytes (PMNs), whereas the latter results in the destruction of periodontal tissues via the activation of osteoclasts and the elevation of matrix metalloproteinase (MMP) concentrations. The overproduction of lipid peroxides, inflammatory mediators, and oxidized proteins from tissue breakdown further leads to the recruitment and activation of phagocytes, particularly macrophages and neutrophils [28]. These host defense cells, through phagocytosis using the oxygen-dependent pathway, induce a ‘respiratory burst’ or ‘oxidative burst’, which is characterized by an increase in oxygen consumption and enhancement of the production of ROS and other metabolic products [29]. ROS have multiple effects on periodontal tissues, which are determined by the redox state. Under the redox equilibrium, ROS may play a key role in the killing of periodontal pathogens. In particular, ROS can drive the disruption of the cellular oxidative environment, which is important for the survival of periodontal microorganisms, predominantly Gram-negative anaerobic or facultative bacteria. Furthermore, ROS may function as the second messenger that regulates signal transduction and transcription factor expression, providing the cytoprotective effects on periodontal tissues. Thus, ROS derived from respiratory burst caused by primary immune responses may represent one of the first lines of defense against the pathogenic microorganisms. Nonetheless, the overproduction of ROS, which comes from the prolonged recruitment and activation of immune cells due to persistent infection or dysregulated inflammation, possibly causes cytotoxic effects and periodontal tissue damage [28]. One of the considerable examples for such dysregulated inflammation is the bi-directional relationship between diabetes mellitus and periodontitis in which diabetes mellitus has an adverse impact on the progression of periodontitis and vice versa. Diabetes mellitus is well known as a chronic metabolic disease characterized by hyperglycemia that subsequently enhances the hyperinflammation and overproduction of oxidants. The consequences of the exacerbation of inflammatory responses further boost the breakdown of periodontal tissues in individuals suffering from both diabetes and periodontitis [9]. Mechanistically, the impact of ROS on the degradation of periodontal tissues may be attributed to its capacity to oxidize vital biomolecules, such as lipids, proteins, and nucleic acids, as well as to regulate the signal transduction and gene transcription involved in immuno-inflammatory responses and cell death. This further produces a vicious circle among the release of ROS and the breakdown of periodontal tissues via the immuno-inflammatory cascade. In addition to the endogenous sources, the periodontal tissues are likely to be exposed to ROS from exogenous stimuli, including heat, trauma, radiation, cigarette smoking, and dental materials used in clinical procedures, such as bleaching agents, among others [28]. In summary, ROS are produced in periodontal tissues from a wide range of both endogenous and exogenous sources under a variety of physiological and pathological conditions (Figure 1). Upon the concentrations, ROS may function as a good or bad weapon in the progression of periodontal disease (Figure 2).

### 3.2. Innate Antioxidant Defense Systems in Periodontal Tissues

Considering the fact that ROS possibly causes a great harm to biological systems, humans have developed three major antioxidant defense systems, including non-enzymatic low-molecular defense system, enzymatic defense system, and repair system, in order to counteract the release as well as the impact of ROS, balancing the redox homeostasis. An antioxidant can be described as “any substance, that when present at low concentrations compared to those of an oxidizable substrate delays or prevents oxidation of that substrate” [30]. The non-enzymatic system, such as uric acid, albumin, glutathione, melatonin, bilirubin, and polyamines, among others, can interrupt and break the chain reactions caused by free radicals. Meanwhile, the enzymatic system, particularly superoxide dismutase (SOD), catalase (CAT), glutathione peroxidase (GPX), glutathione reductase (GSR), and thioredoxin (TRX), can act as inhibitors or scavengers of ROS. For instance, SOD is one of the most significant cellular enzymatic antioxidants, which catalyzes the dismutation of superoxide to the less-reactive species hydrogen peroxide, whereas CAT located in peroxisome efficiently boosts the transformation of hydrogen peroxide into water and oxygen. Collectively, these two antioxidant systems function synergistically and interdependently to maintain the ROS concentration at an optimum level rather than completely eliminate ROS via two major modes of action. The primary mode is referred to as the preventive mode, which functions as the suppressor of free radical generation, the sequestrator of metal ions, and the quencher of singlet oxygen (^1^O_2_) or electronically excited oxygen, which may cause oxidative damage. The secondary mode is described as scavenging ROS or the chain breaking mode, by which the produced ROS are detoxified [29]. However, the equilibrium between ROS and antioxidants is relative in vivo. In fact, the balance may slightly shift toward ROS, leading to oxidative damage at a low degree. This demands a reparative category of the antioxidant defense system, including specific enzymes, proteolytic systems, phospholipases, peroxidases, and acyl transferases, which removes or repairs the impaired biomolecules [31]. Such a category may be interpreted as the third mode of action of the antioxidant defense system. The fundamental knowledge over the biosynthesis and the biological roles of endogenous antioxidants in the whole body has been relatively well presented in the literature [31,32,33].

Endogenous antioxidants are generally present in all body fluids and tissues in humans. Saliva, which is mostly secreted by three pairs of major salivary glands, including submandibular, parotid, and sublingual, as well as by hundreds of minor salivary glands to a lesser extent, is one of the most versatile and important body fluids [34]. Saliva typically comprises 99% water, and the remaining 1% consists of a variety of electrolytes and proteins. Such complexity in salivary constituents is responsible for the wide range of physiological functions attributed to saliva [35]. Remarkably, saliva is crucial for the innate oral immunity. The functions of the salivary defense proteins involved in both innate and acquired oral immunity were well presented. The authors summarized five primary defense networks of salivary proteins in the whole saliva. First, salivary proteins and peptides that bind to bacteria or to oral surfaces or both may be responsible for the microbial agglutination and/or surface exclusion. Second, salivary cationic peptides and lysozyme may be involved in the lysis of microbial membranes. The third and fourth networks may account for the antifungal and antiviral properties, respectively. Finally, the immune regulatory network, including all salivary proteins that function as immune activators or modulators, may be important for the fine regulation of the local action of the mucosal immune system. Importantly, almost all proteins are multi-functional, suggesting their ability to act together to result in an efficient molecular defense network in the oral cavity [36]. In regard to redox biology, saliva can be considered as a key reservoir of both non-enzymatic and enzymatic antioxidants, which also provide three lines of defense, including prevention, detoxification, and reparation. Since excess ROS is likely to be involved in the development of periodontal disease, protection from salivary antioxidants has been the subject of numerous biomedical investigations in the field of oral health. Of significant importance is the antioxidative role of oral peroxidase (OP). The peroxidase activity of human saliva is attributed to salivary lactoperoxidase (LPO) and, to a lesser extent, myeloperoxidase (MPO). LPO is an enzyme found in a variety of human tissues, glands, and secretions, and it is an important element of the non-specific immune response to microorganism challenge. The major action mechanism of LPO is based on the oxidation of thiocyanate ions (SCN^−^) in the presence of hydrogen peroxide to produce hypothiocyanite ions (OSCN^−^), which cause the dysfunction and inhibition of microorganisms by oxidizing the thiol group of amino acid residues of microbial proteins. Specific to the oral cavity, LPO present in saliva is one of the significant constituents maintaining oral health, for instance, by combating the pathogens in periodontal disease. The physiological roles of human salivary LPO were described in detail [37]. While salivary LPO is of salivary glandular origin, MPO present in saliva is produced by neutrophils migrating into the oral cavity via the gingival crevices. Thus, the salivary MPO activity might indicate the infection or inflammation of oral tissues. Similar to LPO, MPO can also catalyze the oxidation of SCN^−^ ions in the presence of hydrogen peroxide to produce much more bactericidal agent, namely hypothiocyanite ions [38]. In our work, the term OP is utilized to indicate the total activity of both peroxidase species, LPO and MPO. OP detoxifies hydrogen peroxide, the major ROS produced by bacterial metabolism, through the peroxidase-catalyzed oxidation of SCN^−^ to yield potent hypothiocyanite ions that exert antimicrobial activity by oxidizing structural and functional components of microbes. In addition to the antimicrobial capacity, hypothiocyanite can also disrupt the production of hydrogen peroxide by oral microorganisms [39]. SCN^−^ is a pseudohalide thiolate existing in secreted biological fluids, such as saliva, plasma, and urine, among others [40]. There is significant variation in terms of SCN^−^ levels amongst such fluids. In a recent review, saliva was reported to exhibit the highest SCN^−^ levels, with a range from 0.5 up to 2 mM, whereas plasma or urine SCN^−^ concentrations were detected at the μM degree [41]. The remarkable difference of SCN^−^ levels in saliva from those in others may be due to the presence of LPO along with other antimicrobial defenses in saliva [42]. Owing to the large pool of salivary SCN^−^, the oral cavity thus provides a favorable environment for oral peroxidase activity in the presence of H_2_O_2_, underlining the importance of the OP/SCN^−^/H_2_O_2_ system in the host defense. On the other hand, saliva is rich in non-enzymatic antioxidants, such as uric acid, albumin, and glutathione, among others. It is well established that uric acid is one of the predominant salivary antioxidants, which is responsible for more than 70% of the total antioxidant capacity of saliva. Uric acid acts as a sweeping agent that eradicates free radicals [43]. In addition to saliva, gingival crevicular fluid (GCF) is also likely to contain a number of antioxidants, which are locally derived from microbiofilm, neutrophils, and crevicular epithelium. The levels of SOD and glutathione represent the most important enzymatic and non-enzymatic antioxidants in GCF, respectively [29]. Taken together, it is clear to postulate that periodontal tissues overcome the oxidative damage caused by exaggerated ROS mainly through the locally endogenous antioxidants (Figure 3). Thus, a deficiency or depletion in such protection may induce oxidative damage and tissue breakdown, eventually leading to the advanced progression of periodontal disease. Indeed, there has been a growing body of evidence consistently indicating a decrease of the antioxidant status in patients with periodontitis in comparison to periodontally healthy individuals despite the diversity of analytical approaches. Many observational studies were comprehensively summarized in a recent review, indicating that the antioxidant status in blood was negatively correlated with the severity of periodontal disease. The more severe the periodontitis, the more powerful the association. Consistently, it has also been confirmed that the total antioxidant capacity was diminished either systemically in plasma or serum, or locally in saliva or GCF, in patients with any forms of periodontal disease [44]. These findings suggest the important roles of antioxidants in the pathogenesis of periodontal disease. Despite the advances of research concerning this scope, there are still gaps that require elucidation. In particular, the paucity of powerful evidence, such as in vivo data, that explores the dynamics of antioxidants during the progression of periodontal disease seems to be a considerable obstacle.

### 3.3. Oxidative Stress and Its Involvement in the Development of Periodontal Disease via the Immuno-Inflammatory Pathway

Description of oxidative stress. As mentioned above, ROS are physiologically generated during mitochondrial oxidative metabolism and other cellular processes. They are mostly believed to be detrimental entities due to their potency to cause oxidative damage to vital biomolecules. Paradoxically, accumulating evidence has shown that ROS at low concentrations also serve as critical signaling molecules in a variety of cellular processes, suggesting their necessity in biological systems. Thus, ROS should be considered as double-edged swords in which the concentration is a key element determining which edge is displayed [45]. Other factors, such as the species of ROS as well as the subcellular location and cellular source of their generation, may also be related to the effects of ROS [46]. In order to control the ROS under an acceptable threshold, the human body is equipped with a defensive network known as intrinsic antioxidants. The maintenance of a physiological level of oxidants, particularly ROS, is greatly important for biological systems through redox signaling. In contrast, an excessive oxidative status due to the overproduction of ROS and/or the deficiency or incompetence of antioxidant defenses may cause damage to the biomolecules that govern living processes, termed oxidative stress [47]. It is widely accepted that oxidative stress is “an imbalance between oxidants and antioxidants in favor of the oxidants, leading to a disruption of redox signaling and control and/or molecular damage” [48]. According to this definition, it is obvious to perceive that oxidative stress is inextricably involved in the disturbance of redox homeostasis, the interruption of signaling pathways, and the impairment of biomolecules, in turn, significantly contributing to the promotion of the aging process as well as the development of numerous diseases [49].

The causal relationship between ROS-mediated oxidative stress and periodontal disease. The causal relationship between oxidative stress mediated by ROS with periodontal disease can be established based on four lines of evidences. Firstly, it has been found that excessive ROS and subsequent oxidative damage are present at the site of injury. A number of studies have directly demonstrated that the levels of reactive oxygen metabolites in the serum of patients with periodontal disease, particularly chronic periodontitis, were significantly higher than those of periodontally healthy individuals [50,51,52]. Furthermore, there have been observations of the destruction of biomolecules induced by ROS as indirect evidence concerning the association between oxidative stress and periodontal disease. Most publications have focused on biomarkers of lipid peroxidation, such as malondialdehyde (MDA). The findings have consistently suggested that patients suffering from periodontal disease display higher levels of lipid peroxidation than periodontally healthy individuals [53]. Besides, the biomarkers of protein damage and DNA damage have been investigated as well. For instance, the levels of 8-hydroxydeoxyguanosine (8-OHdG), a marker of oxidative DNA damage, in subjects with chronic periodontitis have been demonstrated to be significantly elevated in comparison to those in controls with healthy periodontium [53]. Meanwhile, the evidence on protein oxidation was mostly obtained through the elevation of protein carbonyl, which reflects the oxidative damage to proteins. This biomarker has shown high levels in saliva, GCF, and even serum in individuals suffering from advanced periodontal disease [54,55,56]. Secondly, it has been demonstrated that the time course of production of ROS or oxidative damage may occur previously or simultaneously as tissue injury. One representative of this second evidence line is the experimental research that investigated the liberation of ROS (H_2_O_2_) and oxidative DNA damage biomarker (8-OHdG) using rat models with or without application of periodontal pathogens as the experimental groups and control group, respectively. The authors only observed remarkable expression of H_2_O_2_ in the junctional epithelium and the subepithelial connective tissue in periodontitis-induced rats. Moreover, both 8-OHdG expression in gingival fibroblasts and 8-OHdG levels in the plasma of the periodontitis groups were significantly higher than those of the control group. The positive correlation between plasma 8-OHdG and 8-OHdG-positive fibroblasts was demonstrated as well. These findings suggested that ROS were released by gingival fibroblasts in response to pathogens, and the densities and levels of 8-OHdG appeared to reflect the status of periodontal health [57]. Thirdly, experimentally, direct application of ROS at concentrations over a relevant time course to tissues should reproduce similar damage in vivo to that observed in tissues suffering from injury [58]. For instance, a previous study indicated that the intradermal injection of xanthine and xanthine oxidase as a superoxide-generating system in rats may cause significant infiltration of neutrophils in vivo as determined by histological observations. Furthermore, the intravenous administration of superoxide dismutase (SOD) along with the superoxide-generating system was found to suppress the inflammatory responses and inhibit leukocyte infiltration into the treated sites, further supporting the superoxide-dependent accumulation of inflammatory cells in vivo [59]. However, due to the high reactivity and short lifetime of ROS, it is challenging to design a ROS-generating system for in vivo application. Moreover, the direct measurement of free radical generation in an in vivo model detected very low concentrations of ROS [60], making the direct application of ROS more difficult. Thus, almost all existing reports have investigated the direct effects of ROS using in vitro models that may be implicated in tissue damage in vivo. Periodontal disease, especially in advanced stages, is characterized by the degradation and subsequent loss of ligamentous support and alveolar bone, ultimately leading to tooth loss [61]. A number of available data have shown that excessive ROS may cause non-selective damage to the components of the extracellular matrix of periodontal tissues. It has been demonstrated that ROS can damage proteoglycans, hyaluronan, and collagen, leading to the breakdown of connective tissue [58]. Furthermore, although adverse effects of ROS on bone resorption have not been directly investigated yet, they have been determined to probably hinder alveolar bone formation by inhibiting osteoblastic differentiation and promoting osteoclastogenesis [62,63], suggesting their potential role in bone resorption in periodontal disease. Such a role in bone resorption was further supported by the finding that direct exposure of gingival fibroblasts to hydroxy radical and, to a lesser extent, hydrogen peroxide can degrade alveolar bone proteoglycans in vitro [64]. While these findings have important implications for periodontal pathogenesis, the lack of evidence that in vivo levels of ROS production in periodontal tissues cause such damage still needs to be addressed in the future. Finally, it has been documented that the removal or inhibition of ROS generation may subside tissue damage to a degree similar to the antioxidant action in vivo. The typical case of this evidence line comes from the effects of SOD known as an essential scavenger of ROS. By using rat models, SOD was demonstrated to provide therapeutic effects on periodontal pathogen-induced inflammation and to promote the healing process [65]. Similarly, another study using periodontitis-induced Beagle dogs found that adjunctive SOD significantly improved the clinical symptoms and inhibited the inflammation in comparison to mechanical treatment alone [66]. Taken together, the suggestion with respect to the causal role of oxidative stress mediated by ROS in the pathology of periodontal disease can be drawn. Therefore, it is reasonable to postulate that periodontal disease is initially launched by the interaction between microbial challenge and host response but is further progressed by the oxidative stress paradigm via excessive ROS activities and the aberrant inflammatory state. The current work addresses the immuno-inflammatory cascade regulated by oxidative stress in the development of periodontal disease.

The oxidative stress-mediated immunoinflammatory pathway in periodontal disease. Recently, the great interest into the relationship between oxidative stress with immuno-inflammatory responses and inflammation has provided new perspectives regarding the initiation and progression of diseases, particularly chronic conditions [67,68,69]. Considering periodontal disease is defined as a chronic inflammatory condition in the context of this work, the discovery of the role of oxidative stress-mediated inflammation should be considered as important as the detection of the pathogenic microorganisms in periodontal pathology. As mentioned above, the levels of ROS are elevated in periodontal tissues during the primary immune responses following the periodontally microbial challenge via respiratory burst. Albeit their destructive properties can function as an antimicrobial killer to benefit the whole periodontal apparatus, ROS not only harass the pathogens but also non-selectively damage surrounding cells and even tissues in the periodontium. Such periodontal tissue breakdown may enhance the synthesis of proinflammatory cytokines followed by the recruitment and activation of immune cells, leading to abnormal ROS production and subsequent oxidative stress. In this way, an infinite circuit among immuno-inflammatory responses, ROS-mediated oxidative stress occurrence, and periodontal tissue destruction may occur. The burden of ROS in periodontal tissues can be amplified in the presence of risk factors, particularly diabetes mellitus and tobacco smoking, which have been demonstrated to associate with oxidative stress as well [70,71]. Thus, oxidative stress caused by excessive ROS may act as the culprit of chronic inflammation in periodontal tissues, leading to the development of periodontal disease. Now, the more complex matter is the underlying paradigm by which ROS-induced oxidative stress regulates the immuno-inflammatory responses to cause periodontal tissue damage. To our knowledge, there are three major routes relevant to this issue (Figure 4). Nevertheless, the detailed mechanisms are yet to be fully elucidated.

(i) ROS-induced oxidative stress may cause damage at molecular and cellular levels. Aside from the destruction of biomolecules, this route is featured by the oxidative activation of matrix metalloproteinases (MMPs), which are calcium-dependent zinc-containing enzymes involved in the degradation of the extracellular matrix [71]. From the viewpoint of periodontal disease, MMPs are essential proteases that are associated with the destructive process of periodontium. Among members of the MMP family, MMP-2, MMP-8, MMP-9, and MMP-13 are of great importance in the pathology of periodontal disease [72]. MMPs are synthesized as inactive pro-enzymes in the body. During the development of periodontal disease, they can be activated by independent or cooperative proteolytic cascades, leading to widespread tissue destruction and pathological progression. On the other hand, oxidative non-proteolytic MMP activation seems to play an important role as well [73]. Evidence has figured out that ROS are able to activate MMPs in periodontal tissues via direct oxidation. It was found that MMP-2 and MMP-9 were activated in different cell systems, particularly periodontal ligament fibroblasts, by exposure to ROS, such as hydrogen peroxide [74,75]. Besides, other studies indicated that oxidative stress may increase the extracellular matrix turnover mediated by MMP-2, MMP-8, MMP-9, and MMP-13 [76,77]. The pivotal role of MMPs as regulators of periodontal inflammation was well reported in a recent review. Active MMPs are likely involved in the process and cleavage of signaling molecules, such as cytokines, chemokines, and growth factors, among others, thus modulating their biological functions and/or bioavailability, eventually resulting in destructive and prolonged inflammation [78].

(ii) ROS-induced oxidative stress may regulate signaling pathways and transcription factors involved in immuno-inflammatory responses. The predominant route by which ROS-induced oxidative stress involved in aberrant periodontal inflammation is the activation or inhibition of signaling pathways and transcription factors that modulate the immuno-inflammatory responses. First, it has been reported that ROS may activate the NF-κB signaling pathway [79,80]. Such activation may initiate the transcription of a wide range of genes, including proinflammatory cytokines, chemokines, MMPs, and other inflammatory mediators, eventually resulting in breakdown of periodontal tissues via the sustained inflammatory responses and osteoclastic differentiation [81]. In addition, it has been indicated that ROS may trigger the c-Jun N-terminal kinase (JNK) signaling pathway [82]. It is well-known that JNK is a member of a subfamily of mitogen-activated protein kinases (MAPKs), which play crucial roles in signal transduction of extracellular hormones, growth factors, cytokines, bacterial antigens, and environmental stresses, as well as in immune-mediated inflammatory responses [81]. As an oxidative stress-activated protein kinase, JNK can lead to ROS-induced apoptosis, which may play role in the pathology of periodontal disease [83]. Furthermore, ROS-activated JNK can induce the activation of downstream transcription factors, such as c-Jun/AP-1, in turn operating the expression of different genes in response to cytokines [84]. Inhibitors of JNK have also been demonstrated to efficiently hinder the production of proinflammatory mediators [85,86]. The overproduction of ROS and occurrence of oxidative stress not only exacerbates the periodontal damage by promoting the release and aggregation of proinflammatory cytokines and mediators via the activation of signaling pathways but also by the assembly of inflammasomes. The inflammasome is part of the innate immune system that counters microorganisms or stress challenge via the activation of caspase-1 and the induction of inflammation [87]. In regard to periodontal disease, the expression of the NOD-, LRR-, and pyrin domain-containing protein 3 (NLRP3) inflammasome has been demonstrated to be increased in patients with periodontitis compared to periodontally healthy subjects [88,89]. Following the infection of the periodontal pathogen, namely *P. gingivalis*, it was found that the release of IL-1β, IL-6, and IL-18; the gene expression of pro-IL-1β and pro-IL-18; and the activity of caspase-1 in wild-type mice were significantly enhanced in comparison to NLRP3-deficient mice [90]. The activation of caspase-1 is considered as necessary element for the transformation of IL-1β and IL-18 into active cytokines as well as for the onset of a highly inflammatory form of programmed cell death, so-called pyroptosis. Thus, NLRP3 appeared to associate with the production and maturation of proinflammatory cytokines, such as IL-1β and IL-18, and the initiation of pyroptosis, leading to chronic inflammation [91]. There was evidence indicating that ROS may induce the activation of NLRP3 in periodontal tissues [92]. In addition to the activating aspect, ROS-induced oxidative stress has also been shown to inhibit key transcription factors involved in the tolerance of periodontal disease. Of significance, nuclear factor erythroid 2-related factor 2 (Nrf2), a basic leucine zipper transcription factor, is a crucial element that regulates the transcription of a large group of genes of antioxidants and detoxifying enzymes. Under physiological conditions, Nrf2 is bound to Kelch-like ECH-associated protein 1 (Keap1), which functions as a mediator for Nrf2 degradation, thereby maintaining acceptable levels of Nrf2 and preventing unnecessary transcription of antioxidant genes. In contrast, Nrf2 is dissociated from Keap1 during oxidative stress to upregulate Nrf2-associated antioxidants and detoxifying enzymes, providing important protective effects, such as a reduction of inflammatory signaling pathways and oxidative damage in periodontal tissues. However, the downregulation of the Nrf2 pathway followed by the decreased production of antioxidants were observed to associate with the elevated recruitment of oral PMNs and with the advanced progression of periodontitis. These findings suggested that the overwhelmed oxidative stress may blunt the Nrf2 signaling [93].

(iii) ROS-induced oxidative stress may regulate autophagy activities. Autophagy can be defined as a cellular self-degradative process by which long-lived proteins and damaged organelles are recycled to maintain energy homeostasis. Despite the original classification as a form of programmed cell death, autophagy has been widely considered as a survival mechanism to cope with the external challenges [94]. There have been few evidence regarding the role of autophagy induced by ROS in the pathology of periodontal disease; however, they still remain controversial. It may have a protective impact on periodontal tissues through the selective elimination of specific periodontal pathogens, the regulation of immune and inflammatory responses, and the antagonist of apoptosis. Meanwhile, excessive ROS may function as either an inducer or inhibitor of autophagy activities by targeting autophagy-related genes and/or upstream signaling pathways [95]. Owing to the lack of sufficient evidence regarding the interdependence between ROS-mediated oxidative stress and autophagy in the development of periodontal disease, many more investigations about the redox regulation of autophagy involved in this ailment are required.

## 4. Antioxidant Phytochemicals and Periodontal Disease

### 4.1. Overview of the Role of Antioxidant Phytochemicals in the Prevention and Treatment of Periodontal Disease

Periodontal disease is a global health issue due to its popularity all over the world. It was estimated that the prevalence of periodontal disease has increased over decades, and will further advance in upcoming years [96]. Furthermore, a great number of studies have indicated significant associations between periodontal disease and systemic conditions beyond the local consequences of this ailment, causing a dramatic burden for the mankind [3,4]. As a result, the development of approaches that are efficient and affordable for the prevention and treatment of periodontal disease are highly demanded. It is widely accepted that periodontal disease is preventable by maintaining good oral hygiene and a healthy lifestyle. Unfortunately, albeit these preventive modalities are theoretically simple to implement, it is still difficult to routinely practice them at individual and public levels [97]. In fact, oral health professionals are challenged daily by the treatment of this condition and preventing its re-occurrence. In general, professional mechanical therapies, and surgical interventions in severe cases, combined with patient education in regards to oral hygiene and positive behavior are major parts of the standard treatment of periodontal disease [98]. Recently, owing to the new perspectives concerning the regulation of aberrant immuno-inflammatory responses accompanying periodontal disease by oxidative stress, antioxidant-based strategies have appeared to gradually take their place in the fight to against this condition. Although the human body is equipped by a plethora of endogenous constituents that may function as antioxidants, the regimen that can govern oxidative stress and retrieve redox equilibrium, both locally and systemically, also involves the provision of exogenous antioxidants either through dietary or supplement of any forms of antioxidant compound [99]. Exogenous antioxidants may exert their effects by a variety of manners, including synergistically interacting with enzymatic defenses or other endogenous antioxidants, preventing the generation of ROS or scavenging of ROS or sequestering of metal ions, and acting as immune modulators [100]. Otherwise speaking, the consumption of exogenous antioxidants can maintain or re-establish the redox homeostasis directly by the inhibition of ROS generation and impact as well as indirectly by the enhancement of endogenous antioxidant systems and the regulation of immune responses, providing the prophylactic and therapeutic potency in many diseases, particularly chronic and inflammatory conditions. There has been a growing body of evidence indicating the promising effects of exogenous antioxidants in the management of periodontal disease [101,102,103]. However, the underlying mechanisms remain unclear. They may have a beneficial impact on the government of periodontal disease by (1) by controlling the overproduction and unfavorable functions of ROS; (2) diminishing the generation of proinflammatory cytokines, chemokines, and other inflammatory mediators; and (3) promoting the healing process [102]. On the other hand, herbal medicine, or botanical medicine, or phytomedicine, which refers to the utilization of active ingredient parts of plants, or other plant materials, or combinations has been widely accepted to provide preventive and therapeutic benefits since time immemorial [103]. It was estimated that about 80% of the global population employ herbal products for their basic healthcare [12]. These products are preferred over conventional drugs possibly due to their wide biological activities, high safety, and effective cost. The herbal medicine has been demonstrated to possess a huge array of biological properties, such as antimicrobial, antioxidant, and anti-inflammatory effects, suggesting its implications in the control of periodontal disease [104,105]. Moreover, the safety issues regarding the utilization of conventional antioxidant compounds have arisen over time. A number of publications have indicated the side-effects of long-term or high-dose consumption of these substances, such as skin allergies, premature senescence, and even cancer [106,107,108]. This scenario further suggests that antioxidant-specific herbal medicine may become more mainstream in the management of periodontal disease. Of great importance, phytochemicals, or plant chemicals, are bioactive non-nutrient plant compounds have been an important constituent of herbal medicine. They can be employed as folk remedies or as pharmaceutical sorts in contemporary medicine. Herein, the present work specifically addresses the pharmacological activities of the antioxidant phytochemicals in the prevention and treatment of periodontal disease via the oxidative stress pathways.

### 4.2. Classification of Antioxidant Phytochemicals

Phytochemicals cover a wide array of chemical entities that can be categorized as primary and secondary metabolites based on their role in plant metabolism [109]. The former plays important and indispensable roles in the growth and development of plants, whereas the latter acts as defense mechanisms of plants in response to environmental stressors [110]. Secondary metabolites have been determined to be predominantly responsible for the capacity of medicinal plants in the management of various diseases [109]. They can be further divided into a number of classes and subclasses according to their chemical structure and functional groups that may partly account for the respective physiochemical and pharmacological properties per individual phytocompound. However, the definite classification of these secondary metabolites has not been given so far. Structurally, they can be divided into four major classes, including (1) terpenoids with an isoprene molecule as the structural unit, (2) phenolic compounds consisting of at least one aromatic ring attached to one or more hydroxyl groups, (3) nitrogen alkaloids characterized by the presence of at least one atom of nitrogen, and (4) organosulfur compounds or thiols containing sulfur element in their structure [111]. Since oxidative stress appears to play a central role in the sustained immuno-inflammatory responses that accompany various ailments, including periodontal disease, our work only presents the classification of phytochemicals with remarkable antioxidative effects that can be referred to as antioxidant phytochemicals (Figure 5).

Carotenoids. Carotenoids can be described as a group of terpenoids that are responsible for the yellow, orange, and red colors of flowers, leaves, and fruits, in which β-carotene is the most prominent and lycopene is the most efficient among carotenoids. Carotenoids have been well documented as powerful antioxidants based on their ability to directly scavenge singlet molecular oxygen and peroxyl radicals as well as to synergistically interact with other antioxidants, thereby preventing cells and tissues from oxidative damage. They might be associated with the regulation of cellular signaling and redox-sensitive regulatory pathways as well. The unique chemical structures of carotenoids, which include a diverse system of conjugated double bonds, determine the antioxidant activities [112]. Despite their potential, the antioxidative efficacy of carotenoids in biological systems remains controversial since it has been reported that they may exhibit a tendency to lose their antioxidative behaviors depending on a number of factors, such as high concentrations or great oxygen pressure [113]. Furthermore, increasing evidence has also found that the antioxidant activities of carotenoids might shift toward pro-antioxidant activities upon their redox potentials and their interaction within the in vivo environment [114].

Phenolic compounds. Phenolic compounds in plants are a heterogeneous group with more than 10,000 compounds, representing the largest category of phytochemicals [111]. They have been found to be ubiquitous in nature, such as fruits, vegetables, coffee, tea, and cereals, among others, and have been gradually recognized to provide promising benefits for human health due to their wide variety of biological activities. In regard to antioxidative effects, they mainly function as the scavenger of reactive species of oxygen and nitrogen or as the deactivator of metal ions that may interrupt both the initiation and propagation process of oxidative stress. It has been documented that the antioxidant properties of these molecules are likely to associated with their chemical structures [115]. According to the number of phenol units and substituent groups as well as the linkage pattern among phenol units within their molecular structure, polyphenols can be further divided into phenolic acids, flavonoids, and other phenolics. Phenolic acids typically describe the phenolic compounds containing one carboxylic acid group, and they are found in seeds, skins of fruits, and leaves of vegetables, among others. Phenolic acids comprise hydroxycinnamic acids and hydroxybenzoic acids, which are derived from cinnamic acid and benzoic acid, respectively. Some predominant representatives of hydroxycinnamic acids include ferulic acid, caffeic acid, and *p*-coumaric acid, whereas gallic acid and ellagic acid are common compounds in the hydroxybenzoic acid group. While the mechanisms involved in the antioxidant activity of phenolic acids remain unclear, radical scavenging via hydrogen atom donation due to the reactivity of the phenol moiety has been proposed as the crucial route. Moreover, the substituents on the aromatic ring may affect the stabilization of phenolic acids, which possibly determines their radical-quenching capacity [116]. Flavonoids are one of the most studied families, with approximately 4000 different structures known [111]. They are composed of flavonols (quercetin, kaempferol), flavones (chrysin, luteolin, apigenin), isoflavones (genistein), flavanols (catechin, epicatechin, epicatechin gallate, epigallocatechin gallate), flavanones (hesperidin), and anthocyanins (cyanidin, pelargonidin) [117,118]. All of these subgroups share the same diphenylpropane skeleton. They can act as reducing agents, hydrogen donators, singlet oxygen quenchers, superoxide radical scavengers, and even metal chelators. Furthermore, they have been indicated to activate antioxidant enzymes, detoxify α-tocopherol radicals, inhibit oxidases, alleviate nitrosative stress, as well as elevate the levels of uric acid and other low-molecular-weight molecules. The antioxidant properties conferred on flavonoids may come from the phenolic hydroxyl groups attached to ring structures [117]. Based on the body of published evidence indexed on PubMed up to 2019 regarding the most studied phytochemicals as related to oxidative stress, a wide variety of phenolic compounds with antioxidant capacity were reported, including anthraquinones, coumarins, curcumin, stilbenes (resveratrol), tannins, and xanthones [119]. Every compound probably has different modes of actions corresponding to its structural specificity but consistently confers the antioxidative efficacy. Similar to carotenoids, there is the suspicion that phenolic compounds may act as double-edged molecules in the maintenance of redox equilibrium depending on their concentrations and other elements [120,121].

Others. While carotenoids and polyphenols represent two main kinds of antioxidant phytochemicals, a large number of other phytochemicals has been demonstrated to possess antioxidant capacity. Of importance is organosulfur compounds, which are generally found in cruciferous vegetables, such as broccoli, cauliflower, brussel sprouts, garlic, and onion. Among them, the *Allium* genus of flowering plants, including garlic and onion, contains important compounds, such as cysteine sulfoxides, which have four types: alliin, methiin, propiin, and isoalliin. Alliin, which is rich in garlic, is converted into allicin through enzymatic hydrolysis when garlic is crushed during chewing or cooking [122]. Existing evidence has demonstrated the powerful antioxidant properties of allicin-derived compounds, providing protection against various chronic diseases [123]. Allicin may prevent cells from oxidative stress via induction of the generation of glutathione and antioxidative allicin derivatives, thereby reducing the levels of ROS and scavenging free radicals [124]. Besides, the antioxidant effects of the organosulfur compounds may be attributed to the chelation of metals and the induction of the antioxidant response pathway mediated by Nrf2 [125]. In addition to organosulfur compounds, several studies have also reported the antioxidant activities of alkaloid fractions derived from numerous plants [126,127,128]; however, the results are inconsistent and inconclusive. Importantly, the mechanisms related to their antioxidant properties remain unknown.

### 4.3. Pharmacological Activities of Antioxidant Phytochemicals in the Prevention and Treatment of Periodontal Disease via the Inhibition of Oxidative Stress Pathways

Conventional preventive and therapeutic modalities in the management of periodontal disease mainly involve the maintenance of good oral hygiene and healthy habits as well as the execution of professionally non-surgical and surgical approaches in order to eliminate the etiological microorganisms and modifiable risk factors. In addition, adjunctive medications, such as antibiotics and anti-inflammatory drugs, may function as additional counteracts to microbial challenge and pathological inflammation. The clinical benefits of these adjuncts in the control of periodontal disease are yet consistent and conclusive, especially concerning their significant unwanted side effects. Thus, the accessory resolution of the infection/immune-inflammation cascades via another route, i.e., oxidative stress, may be promising and practical in regard to periodontal disease. The current review mainly focuses on the pharmacological activities of compounds belonging to two major categories of antioxidant phytochemicals, carotenoids and polyphenols, thereby discussing their role in the inhibition of oxidative stress pathways, both upstream and downstream, that appears to lie in the core of the inflammation underlying the development of periodontal disease (Figure 6). Albeit the effectiveness of these entities has not been sufficiently explored yet, it has raised interest in their potential implications as prophylactic and therapeutic adjuncts for periodontal disease.

Anti-periodontally microbial activity. As mentioned above, periodontal disease is likely initiated by microbial challenge derived from microbial biofilm and further progressed by host responses, including the overproduction of ROS via respiratory burst and other processes. Thus, the elimination of periodontally pathogenic microorganisms can be considered as the first step in the management of periodontal disease. Existing evidence has suggested the potential of polyphenols to fight against such pathogens. A comprehensive in vitro assay, which investigated the inhibitory effects of 48 polyphenolic compounds on periodontal bacteria growth and biofilm formation, found that curcumin was the most potent inhibitor, followed by pyrogallol, pyrocatechol, and quercetin [129]. Consistently, common dietary sources of polyphenols, such as blueberry extract and tea polyphenols, have also been demonstrated to hinder bacterial growth and biofilm formation in other experiments [130,131]. These findings have been further supported by a few human clinical studies. For instance, the antimicrobial efficacy of the local delivery of a gel containing 1% curcumin was examined in 25 patients diagnosed with chronic periodontitis by using a split mouth design in which two sites in the contralateral quadrants with a similar probing pocket depth at baseline were selected in every patient. Then, the treated sites received the gel along with scaling and root planning, whereas the contralateral sites or control sites received scaling and root planning alone. The results showed that the topical application of 1% curcumin gel provided a significant reduction in the presence of microbes at the treated sites compared to those at the control sites, suggesting the potential of curcumin in the inhibition of periodontal microbial growth [132]. The mechanisms of the antibacterial action of such compounds, particularly in regard to periodontal pathogens, have not been fully investigated yet. In general, these compounds have been proposed to involve many sites of action at cellular levels. For instance, owing to their lipophilic character, phenolic compounds can interact with the bacterial cell membrane, leading to damage of the cytoplasmic membrane, the coagulation of cellular content, and the inhibition of intracellular enzymes, thereby exerting antimicrobial activity [133,134]. Moreover, their antibacterial properties may be enhanced through several mechanisms, such as the alteration of membrane permeability, the inhibition of nucleic acid synthesis, and the interruption of energy metabolisms, among others [135]. Further investigations are highly required to clarify the mechanisms underlying the antibacterial activity of the antioxidant phytochemicals on periodontally pathogenic microorganisms.

Anti-oxidative stress activities. Antioxidant phytochemicals have been shown, both in vitro and in vivo, to have beneficial effects on the management of periodontal disease by dealing with oxidative stress and re-establishing the redox balance. As mentioned above, the well-being of periodontium is probably dependent on the steady liberation of physiological concentrations of reactive species, such as superoxide, hydrogen peroxide, hydroxyl radicals, nitric oxide, and hypochlorous acid, to an extent. Phyto-antioxidants, such as polyphenolic compounds in green tea (catechin), including catechin, gallocatechin, epicatechin, epigallocatechin, epicatechin gallate (ECG), and epigallocatechin gallate (EGCG), have been well known to inhibit the generation of ROS and lysosomal enzymes as well as to scavenge ROS, suggesting the plausible role of green tea as an antioxidant in relation to periodontal disease [136]. Besides, the use of resveratrol, a polyphenol compound that is found in grapes and wine, was reported to diminish the production of nitric oxide in a concentration- and time-dependent manner using a human periodontal ligament cell model stimulated by LPS of the periodontal pathogen, namely *P. gingivalis* [137]. Previously, luteolin, quercetin, and genistein, which are members of the flavonoids class, were also determined to be potent inhibitors of nitric oxide synthesis using an LPS-exposed human gingival fibroblast model [138]. The antioxidant phytochemicals not only inhibit the generation and/or side effects of reactive species but also promote the antioxidant status through the interaction with enzymatic defense. The consumption of tomatoes containing natural lycopene has been found to enhance the levels of antioxidants in plasma [139,140]. Remarkably, another study was unable to demonstrate either beneficial or adverse effects of β-carotene, lutein, or lycopene supplementation on oxidative stress in healthy individuals, suggesting that carotenoid supplementation may fail to significantly improve antioxidant defense under normal conditions [141]. In regard to polyphenols, it was found that green tea significantly increased the activity of an endogenous antioxidant, namely glutathione-S-transferase, subsequently decreasing the severity of gingival inflammation and increasing the periodontal parameters in patients with chronic periodontitis [142]. Similarly, another study on periodontitis patients reported that the treatment group that took dark chocolate, a rich source of flavonoids, may increase the total antioxidant capacity and decrease lipid peroxidation in comparison to the control group that took white chocolate [143].

Immune regulatory and anti-inflammatory activities. Due to the bi-directional relationship between oxidative stress and immuno-inflammatory responses, the immune regulatory and anti-inflammatory activities should not be overlooked in order to inhibit the oxidative stress pathways. There has been accumulating evidence regarding such activities induced by antioxidant phytochemicals in regards to periodontal disease. It was found that β-carotene may potentially inhibit the production of inflammatory cytokines, such as tumor necrosis factor α (TNF-α) and interleukin 6 (IL-6), via NF-κB signaling without cell damage using THP-1 monocytes stimulated by LPS of *P. gingivalis* [144]. Furthermore, higher consumption of β-carotene was shown to improve the inflammatory responses, both locally and systemically, in non-smokers with chronic periodontitis [145]. On the other hand, a few in vitro and in vivo studies have provided evidence on the potential effects of polyphenols on the alleviation of inflammation and damage associated with periodontal disease. One in vitro experiment using the three-dimensional co-culture model of gingival epithelial cells and fibroblasts indicated that green tea polyphenol epigallocatechin-3-gallate and cranberry proanthocyanidins may synergistically act with the human antimicrobial peptide cathelicidin to reduce the LPS-stimulated release of cytokines [146]. By using the rodent models, it has been demonstrated that the polyphenol treatment may ameliorate the in vivo inflammation and damage associated with periodontal disease. For instance, *Hypericum perforatum*, which is a medicinal plant species containing a wide variety of polyphenolic compounds, was determined to exert potent anti-inflammatory effects in periodontitis-induced rats, and, in turn, significantly improved the hallmarks of immune inflammation, including neutrophil infiltration and other inflammatory cells’ recruitment, cytokine production, and inflammatory mediator expression, such as NF-κB, among others [147]. Another recent study also reported that the oral intake of the green tea polyphenol epigallocatechin-3-gallate may downregulate the expression of *P. gingivalis*-induced inflammatory mediators, such as IL-1β, IL-6, and TNF-α, probably mitigating subsequent damage in periodontitis mice [148].

Wound healing-promoting activities. The plausible interplay between oxidative stress and tissue breakdown may highlight the benefits of wound healing-promoting activities in the management of periodontal disease. A growing body of evidence, including experimental and clinical studies, has indicated those activities of antioxidant phytochemicals. Since alveolar bone resorption is a predominant hallmark of the progression of periodontal disease, the majority of existing literature has focused on the anti-osteoclast or bone resorption regulatory effects. One of the main carotenoids, β-cryptoxanthin, was found to inhibit alveolar bone resorption in human periodontal ligament cells [149]. Similarly, A-type cranberry proanthocyanidins, a form of polyphenols, was also determined to disrupt the maturation and physiology of osteoclastic cell even at a low concentration of 10 μg/mL, preventing bone resorption [150]. These in vitro findings have been further supported by in vivo experiments. The administration of a keto-carotenoid, namely astaxanthin, was postulated to prevent alveolar bone loss by increasing osteoblastic activity and decreasing osteoclastic activity in periodontitis-induced rats [151]. The same results have been observed in a family of polyphenolic compounds, such as myricetin [152], mangiferin [153], and resveratrol and/or curcumin [154], in periodontitis-induced rodent models. Remarkably, significant improvement of periodontal healing has also been observed in the treatment group using polyphenols compared to the control group in several clinical studies [155,156,157].

### 4.4. Challenges and Future Prospects

Albeit antioxidant phytochemicals have been revealed to possess many favorable pharmacological activities associated with periodontal disease, there are many gaps requiring further investigations. An important point to consider is the low bioavailability of these compounds in most cases [158,159]. In regard to the human body, the term ‘bioavailability’ can be defined as substances obtained from ingestion that reach the circulatory system for further transportation to target tissues and subsequent exacerbation of biological functions [160]. Thus, for a successful intervention, the issue of optimizing the bioavailability of antioxidant phytochemicals, such as the development of user-friendly delivery systems, should be investigated. On the other hand, the safety issue has always been of great concern over time. The possibility that natural compounds may exert lower biological effects than their synthetic counterparts implies the demand of using the antioxidant phytochemicals in larger amounts, leading to the risk of overdose. As mentioned above, the double-edged effect not only involves ROS but also antioxidants. The high concentrations of supplemental exogenous antioxidants may lead to more severe disruption of the redox equilibrium rather than maintenance. Therefore, further research regarding the toxicity and side effects of antioxidant phytochemicals upon their dosage is highly required. Besides, the detailed mechanisms of action underlying the mentioned pharmacological activities of antioxidant phytochemicals in the management of periodontal disease also need to be investigated in the future.

## 5. Conclusions

Despite the abundance of possible mechanisms involved in the periodontal disease, oxidative stress has become an area of intensive research in regards to the immuno-inflammatory paradigm underlying this ailment. Unveiling such an oxidative pattern of periodontal pathology is therefore crucial to the development of antioxidant-based strategies in the prevention and treatment of periodontal disease. Antioxidant phytochemicals have been demonstrated to function as prophylactic and therapeutic agents for this condition. The wide array of pharmacological activities of these entities provide promising opportunities to develop novel antioxidant modalities, which function not only as powerful anti-oxidative stress elements but also as efficient antimicrobial, immune regulatory, anti-inflammation, and wound healing-promoting agents. Further investigations regarding the underlying mechanisms for such activities are highly required. Moreover, despite the potency of antioxidant phytochemicals as an adjunct to conventional approaches, there are still voids with respect to the bioavailability and dosage of these compounds for clinical application in periodontology. Future research to explore the detailed mechanism of oxidative stress-regulated immuno-inflammatory cascades as well as to overcome the shortcomings of antioxidant phytochemical-based remedies is warranted in the management of periodontal disease.

## Figures and Tables

**Figure 1 antioxidants-09-01211-f001:**
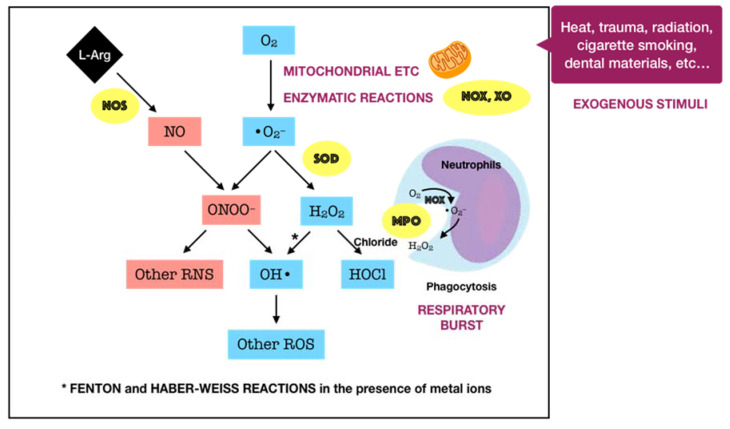
Schematic representation of the generation of major reactive species in periodontal tissues. In general, the main generator of ROS is the mitochondrial electron transport chain ETC, liberating superoxide, which further leads to the cascade of other ROS production. Besides, enzymatic reactions catalyzed by NADPH oxidase (NOX) or xanthine oxidase (XO) also significantly participate in ROS production. Upon the invasion of periodontal bacteria, ROS generation in periodontal tissues is enhanced via respiratory burst following the phagocytosis by immune cells, such as neutrophils. Of significance is the production of hypochlorous acid (HOCl) from hydrogen peroxide and chloride in a reaction catalyzed by myeloperoxidase (MPO) derived from neutrophils. On the other hand, nitric oxide (NO), which is product of the metabolism of L-arginine via the catalysis of nitric oxide synthase (NOS), is responsible for the production of another group of reactive species, namely reactive nitrogen species (RNS). Importantly, peroxynitrite (ONOO^−^), which is produced through the reaction between NO and superoxide, is not only a powerful RNS but also represents the merging of the ROS and NO pathways. In addition to internal sources, periodontal tissues are probably exposed to ROS from external stimuli as well.

**Figure 2 antioxidants-09-01211-f002:**
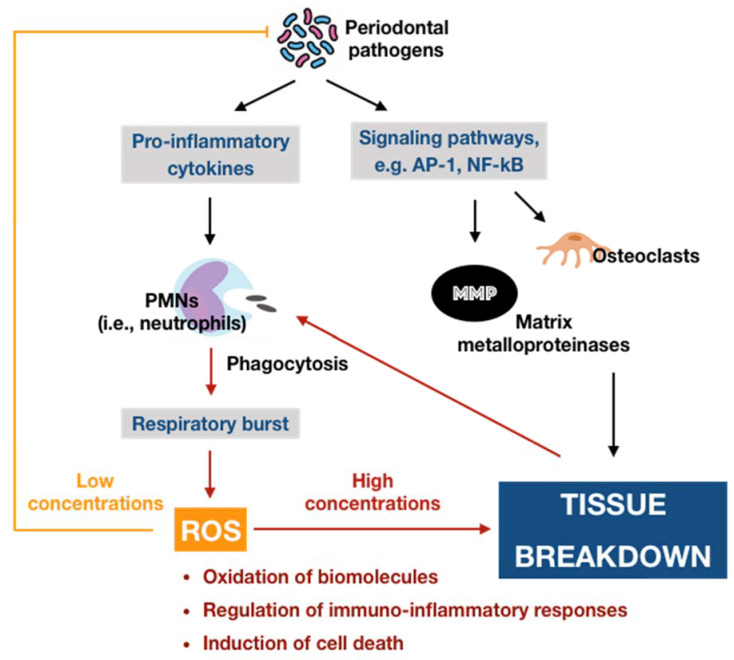
Schematic representation of the double-edged effects of ROS in periodontal disease. At physiological levels, ROS may contribute to the killing of periodontal pathogens and function as the second messenger that mediates biological processes, providing cytoprotective effects. In contrast, excessive ROS can induce many adverse effects, generating a vicious circle among ROS and tissue breakdown via the immuno-inflammatory cascade.

**Figure 3 antioxidants-09-01211-f003:**
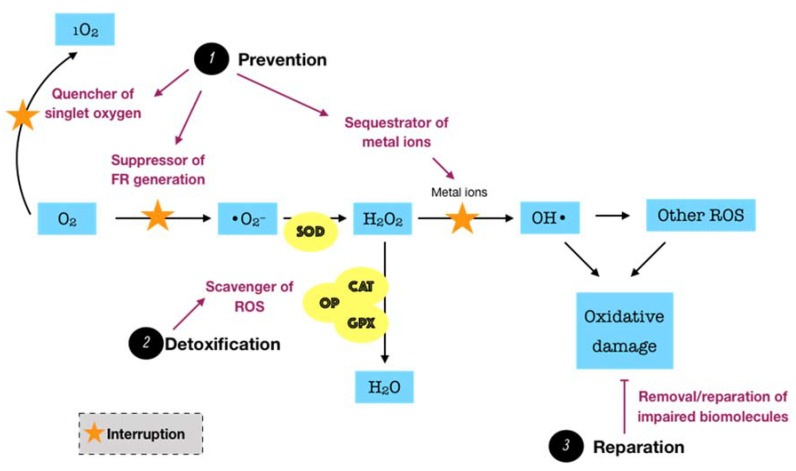
Schematic representation of the functions of endogenous antioxidant defense systems. Endogenous antioxidants, which present in all body fluids and tissues, consist of enzymatic and non-enzymatic systems. They counteract ROS and maintain redox balance via three major modes of action. First, the preventive mode (prevention) functions as the quencher of singlet oxygen, suppressor of free radical (FR) production, and sequestrator of metal ions. Second, the detoxifying mode (detoxification) involves the scavenger of ROS or interruption of chain reactions. Third, the reparative mode (reparation) functions to remove or repair the biomolecules suffering from oxidative damage. In the oral cavity, saliva and gingival crevicular fluid represent important local reservoirs of endogenous antioxidants.

**Figure 4 antioxidants-09-01211-f004:**
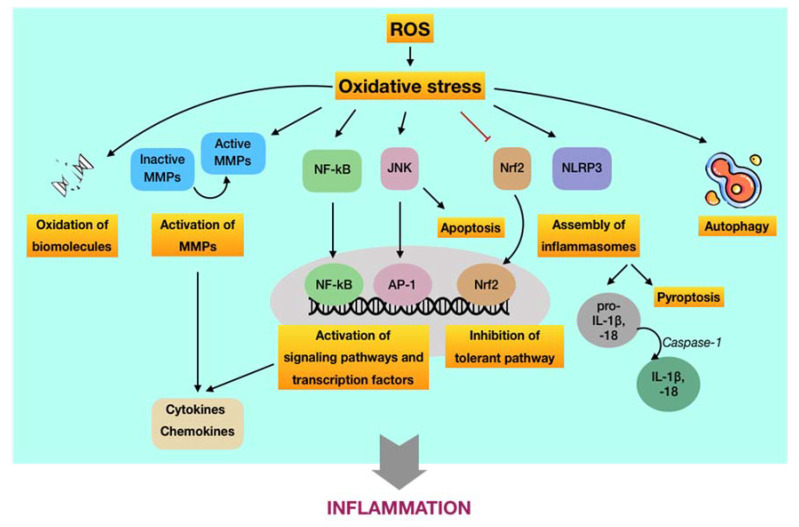
Schematic representation of the underlying pathways by which ROS-induced oxidative stress regulates inflammation, contributing to the progression of periodontal disease. First, oxidative stress may induce oxidation of vital biomolecules and activate matrix metalloproteinases MMPs, which are regulators of inflammation. Second, oxidative stress may enhance the production and expression of proinflammatory cytokines, chemokines, among others, through the activation of NF-κB-, JNK-, and NLRP3-dependent pathways. Moreover, the JNK route may induce apoptosis, whereas the NLRP3 route may cause pyroptosis. Besides, severe oxidative stress may also hinder the tolerant mechanism via suppression of the Nrf2 paradigm. Finally, oxidative stress may function as an inducer or inhibitor of autophagy activities.

**Figure 5 antioxidants-09-01211-f005:**
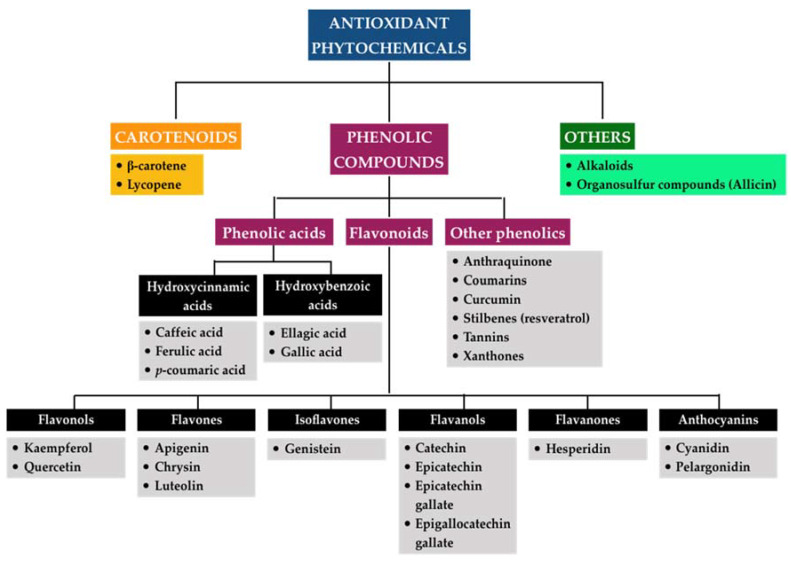
Classification of antioxidant phytochemicals depending on their chemical structure and functional groups. There are two main categories, including carotenoids and phenolic compounds. Phenolic compounds can be further divided into subgroups, which are phenolic acids, flavonoids, and other phenolics. Phenolic acids consist of hydroxycinnamic acids and hydroxybenzoic acids, whereas flavonoids are composed of flavonols, flavones, isoflavones, flavanols, flavanones, and anthocyanins. Other phenolics are a heterogenous group in which anththraquinone, coumarins, curcumin, stilbenes, tannins, and xanthones are representatives. In addition, entities with different chemical structures belonging to alkaloids and organosulfur compounds are grouped together under one category. Every compound may have different mode of actions according to their specific structure but consistently exert remarkable antioxidative efficacy.

**Figure 6 antioxidants-09-01211-f006:**
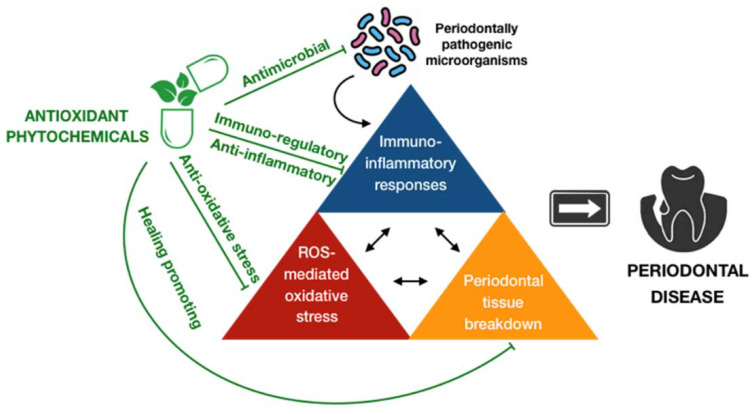
Schematic representation of the potential preventive and therapeutic role of antioxidant phytochemicals in the management of periodontal disease. Periodontal disease is initially caused upon the microbial challenge, then further progressed by the interplay among oxidative stress and immuno-inflammatory responses and tissue damage. Antioxidant phytochemicals seem to be capable of inhibiting oxidative stress pathways, both upstream and downstream, thereby controlling the progression of periodontal disease. There are four main pharmacological activities involved in such potency of these entities, including anti-microbial, anti-oxidative stress, immune regulatory and anti-inflammatory, and wound healing-promoting activities.

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
