# Peer review of "The Promising Role of Antioxidant Phytochemicals in the Prevention and Treatment of Periodontal Disease via the Inhibition of Oxidative Stress Pathways: Updated Insights"

_antioxidants, 2020, doi:10.3390/antiox9121211_

Round 1

Reviewer 1 Report

The topic of this review concerns the role of oxidative stress mechanisms in the development of periodontal disease and the helpful effects of phytochemicals in prevention and treatment of this disease.

The importance of saliva components in innate defense systems is expressed in subchapter 3.2. It is necessary to describe in more detail specific conditions of immune defense in saliva. The role of lactoperoxidase (also known as salivary peroxidase) should by extended as well as the role of this enzyme in production of the bactericidal agent hypothiocyanite. Indication of thiocyanate level in saliva in contrast to serum level should also be helpful.

In lines 260-262, authors state that oral peroxidase is a salivary enzyme consisting of 80 % salivary peroxidase and 20 % myeloperoxidase without giving a reference for this wrong statement. The latter enzyme is not produced in salivary glands. It is released from neutrophils. The presence of myeloperoxidase in saliva is therefore associated with inflammatory processes.

Considering the ion composition of saliva, it is unlikely that a chloride oxidation by myeloperoxidase should occur.

Chapter 4.           The role of phytochemical antioxidants in periodontal disease is described very superficial. Flavonoids are well known to affect catalytic cycles in lactoperoxidase and myeloperoxidase and to modulate the (pseudo)halogenating activity of these enzymes. These data are not discussed.

There is an inappropriate mixing between the sequential one-electron reduction of O2 to H2O and the four-electron reduction of O2 to H2O by cytochrome c oxidase in mitochondria. In the latter process, redox intermediates such as O2·-, H2O2, and ·OH are not involved.

Fig. 1.    Peroxynitrite is directly linked with HOCl by myeloperoxidase. This is wrong. HOCl is an uncharged molecule, and not as indicated an ion.

Fig. 3.    What is active oxygen? This term is not specified.

Line 354               replace hyaluronic by hyaluronan; collagen is not a proteoglycan

There are numerous misprints throughout the manuscript.

Author Response

Manuscript ID: Antioxidants-999518

Dear Reviewer,

Thank you very much for your critical comments on our manuscript. You pointed out the major issues requiring further modifications, thereby improving the scientific content in our manuscript. We have revised our manuscript according to your comments. Please refer to the list of Responses to Comments in the below pages.

We hope that the revised manuscript will be satisfactory for publication in the Antioxidants, and look forward to hearing from you about the acceptability of the paper. Thank you so much for your considerations.

With best regards,

I-Ta Lee, Ph.D.

School of Dentistry, College of Oral Medicine, Taipei Medical University, Taipei, Taiwan

250 Wuxing St. Taipei 11031, Taiwan

Tel: +886-2-27361661 ext. 5162

Fax: +886-2-27362295

E-mail addresses: itlee0128@tmu.edu.tw

  1. The importance of saliva components in innate defense systems is expressed in subchapter 3.2. It is necessary to describe in more detail specific conditions of immune defense in saliva. The role of lactoperoxidase (also known as salivary peroxidase) should be extended as well as the role of this enzyme in production of the bactericidal agent hypothiocyanite. Indication of thiocyanite level in saliva in contrast to serum level should also be helpful.

Responses to Comments:

Thank you very much for your critical comments. We added a brief description of the role of saliva in innate immune defense (Line 380-395). The role of lactoperoxidase was expanded as well (Line 401-409). We mentioned and cited the review papers regarding these points in the manuscript for further detailed reading if needed.

  1. In lines 260-262, authors state that oral peroxidase is a salivary enzyme consisting of 80% salivary peroxidase and 20% myeloperoxidase without giving a reference for this wrong statement. The latter enzyme is not produced in salivary glands. It is released from neutrophils. The presence of myeloperoxidase in saliva is therefore associated with inflammatory processes.

Responses to Comments:

Thank you very much for your critical comments. The statement “oral peroxidase is a salivary enzyme consisting of 80% salivary peroxidase and 20% myeloperoxidase” was written based on following references, [doi:10.5539/gjhs.v6n5p87; DOI: 10.1016/j.freeradbiomed.2003.08.001]. We did not intend to describe myeloperoxidase is produced in salivary glands. In order to avoid misunderstanding, we modified this statement (Line 400-401). The description of oral peroxidase was adjusted as well (Line 400-423).  

  1. Considering the ion composition of saliva, it is unlikely that a chloride oxidation by myeloperoxidase should occur.

Responses to Comments:

Thank you very much for your critical comment. To our knowledge, chloride is presented in saliva composition [doi: 10.15713/ins.ijcdmr.121; DOI: 10.1111/joor.12664]. Thus, we think chloride oxidation by myeloperoxidase can occur in saliva. The production of HOCl from myeloperoxidase/H2O2-dependent oxidation of chloride anion in periodontal disease was documented [DOI: 10.1016/S1991-7902(09)60008-8].

  1. Chapter 4. The role of phytochemical antioxidants in periodontal disease is described very superficial. Flavonoids are well known to affect catalytic cycles in lactoperoxidase and myeloperoxidase and to modulate the (pseudo)halogenating activity of these enzymes. These data are not discussed.

Responses to Comments:

Thank you very much for your critical comments. Our review mainly focused on the immuno-regulatory pathway regulated by oxidative stress in periodontal pathology. Based on such pathway, we collected available findings and summarized into four major pharmacological activities belonging to antioxidant phytochemicals. Our purpose was to provide evidences suggesting the role of antioxidant phytochemicals in the inhibition of the oxidative stress pathways, both upstream and downstream, that appears to lie in the core of the inflammation underlying the development of periodontal disease. The effectiveness and underlying mechanisms remain unclear. We mentioned this limitation in the section 4.4 (Line 1020-1023). We hope the precise mechanisms of action of these phytochemicals related to periodontal disease will be addressed in the future.

  1. There is an inappropriate mixing between the sequential one-electron reduction of O2 to H2O and the four-electron reduction of O2 to H2O by cytochrome c oxidase in mitochondria. In the latter process, redox intermediates such as •O2−, H2O2, and OH• are not involved.

Responses to Comments:

Thank you very much for your critical comments. In the context of this work, we described the sequential univalent reduction of O2 in mitochondrial ETC to generate ROS intermediates rather than the tetravalent reduction by cytochrome c oxidase (Line 147-159). Besides, enzymes involving in ROS generation were presented as well (Line 229-235). The word “COX” indicated cyclooxygenases, not cytochrome c oxidase. We also modified the Figure 1 (Line 291) to avoid misunderstanding.

  1. Fig. 1. Peroxynitrite is directly linked with HOCl by myeloperoxidase. This is wrong. HOCl is an uncharged molecule, and not as indicated an ion.

Responses to Comments:

Thank you very much for your critical comments. These typographical errors were adjusted in Figure 1 (Line 291). The production of HOCl from hydrogen peroxide and chloride in a reaction catalyzed by the enzyme myeloperoxidase derived from neutrophils was corrected. HOCl was corrected into uncharged molecule as well.

  1. Fig. 3. What is active oxygen? This term is not specified.

Responses to Comments:

Thank you very much for your critical comment. The non-specific word “active oxygen” was corrected into the term “singlet oxygen” in the text (Line 369-370; Line 450) and in Figure 3 (Line 446).

  1. Line 354 replace hyaluronic by hyaluronan; collagen is not a proteoglycan.

Responses to Comments:

Thank you very much for your critical comments. These typing errors were modified. “Hyaluronic” should be written as “hyaluronic acid” or “hyaluronan”. We corrected this word into “hyaluronan” (Line 537) as the suggestion. We did not intend to describe collagen as proteoglycan. We modified the related statement in order to avoid the misunderstanding (Line 537-538).

  1. There are numerous misprints throughout the manuscript.

Responses to Comments:

Thank you very much for your critical comment. The misprints in our manuscript were checked and corrected. The modifications were highlighted using “Track Changes” function.

Reviewer 2 Report

This is an interesting review addressing a topic of major relevance in the field of the promising health effects of antioxidant phytochemicals in the contest of periodontal disease.  Although the review is well organized and easy to read, the authors may wish to address the following points for clarification or corrections.

Please add in paragraph 3.1 a general explanation about the beneficial and deleterious effect of nitric oxide (NO) production. Also explain the different isoforms and localization of nitric oxide synthase (NOS) enzyme. It is important to explain when inducible form of NOS is activated, that NOS could be the target of ROS and it can switch its activity by producing ROS instead of NO.

Lines-348-350 « Thirdly, experimentally direct application of ROS at in vivo concentrations over a relevant time course to tissues has been shown to reproduce similar damages to that observed in tissues suffering from injury ». Please

-add the reference

-clarify how authors applicate ROS directly

-explain which are in vivo concentration of ROS

Lines 624-643, paragraph about « anti-periodontally microbial activity »

Please explain which are the mechanisms allowing the decreasing of bacteria growth upon polyphenol compound treatment. Specifically, details the mechanisms of dietary sources of polyphenols, such as blueberry extract and tea polyphenols, as well as those of curcumin.

Please explain the origin of NO upon LPS stimulation in the experiments describing the use of resveratrol (lines 653-656).

Suggestions:

Line 231 change “provoked” into “developed”

Line 310 change “provocation” into “status”

Line 342 change “expression” into “production”

Lines 418 and 642 change “prohibition” into “inhibition”

Line 450 change “prohibit” into “inhibit”

Line 654 change “expression” into “production”

Line 655 change “dose” into “concentration”

Author Response

Manuscript ID: Antioxidants-999518

Dear Reviewer,

Thank you very much for your critical comments on our manuscript. You pointed out the major issues requiring further modifications, thereby improving the scientific content in our manuscript. We have revised our manuscript according to your comments. Please refer to the list of Responses to Comments in the below pages.

We hope that the revised manuscript will be satisfactory for publication in the Antioxidants, and look forward to hearing from you about the acceptability of the paper. Thank you so much for your considerations.

With best regards,

I-Ta Lee, Ph.D.

School of Dentistry, College of Oral Medicine, Taipei Medical University, Taipei, Taiwan

250 Wuxing St. Taipei 11031, Taiwan

Tel: +886-2-27361661 ext. 5162

Fax: +886-2-27362295

E-mail addresses: itlee0128@tmu.edu.tw

  1. Please add in paragraph 3.1 a general explanation about the beneficial and deleterious effect of nitric oxide (NO) production. Also explain the different isoforms and localization of nitric oxide synthase (NOS) enzyme. It is important to explain when inducible form of NOS is activated, that NOS could be the target of ROS and it can switch its activity by producing ROS instead of NO.

Responses to Comments:

Thank you very much for your critical comments. We added a fundamental description of NO, including its double-edged effects as well as its production via three isoforms located at different locations (Line 171-187). A few comprehensive reviews regarding these points were mentioned and cited for further detailed reading if needed. Importantly, the interplay between ROS and NO pathways as well as the shifting from NO production into ROS generation by inducible NOS regulated by ROS was introduced too (Line 188-205). 

  1. Line 348-350 “Thirdly, experimentally direct application of ROS at in vivo concentrations over a relevant time course to tissues has been shown to reproduce similar damages to that observed in tissues suffering from injury”. Please add the reference; clarify how authors applicate ROS directly; explain which are in vivo concentration of ROS.

Responses to Comments:

        Thank you very much for your critical comments. The statement “Thirdly, experimentally direct application of ROS … tissues suffering from injury” was modified into “Thirdly, experimentally direct application of ROS at concentrations over a relevant time course to tissues should reproduce similar damages in vivo to that observed in tissues suffering from injury” to avoid misunderstanding (Line 520-522). We added reference for this statement. We also expanded the discussion related to this point (Line 522-533), and pointed out the paucity of in vivo information (Line 544-546). 

  1. Line 624-643, paragraph about “anti-periodontally microbial activity”. Please explain which are the mechanisms allowing the decreasing of bacteria growth upon polyphenol compound treatment. Specifically, details the mechanisms of dietary sources of polyphenols, such as blueberry extract and tea polyphenols, as well as those of curcumin.

Responses to Comments:

Thank you very much for your critical comments. Our review mainly focused on the immune-regulatory pathway regulated by oxidative stress in periodontal pathology. Next, our purpose was to collect available findings, then summarizing into four major pharmacological activities, including anti-periodontally microbial activity, belonging to antioxidant phytochemicals, thereby suggesting their role in the inhibition of the oxidative stress pathways, both upstream and downstream, that appears to lie in the core of the inflammation underlying the development of periodontal disease. The underlying mechanisms remain unclear. Thus, we added some possible mechanisms as proposed by previous reviews (Line 917-927). We hope the precise mechanisms of action of these phytochemicals will be addressed in the future.

  1. Please explain the origin of NO upon LPS stimulation in the experiments describing the use of resveratrol (lines 653-656).

Responses to Comments:

        Thank you very much for your critical comment. The authors did not explain the origin of NO upon LPS stimulation in their experiments, so we did not mention this point in our manuscript. To our knowledge, LPS is a major component of the outer membrane of gram-negative bacteria, including P. gingivalis. It can bind to proteins on the cell surface, then present to TLR4 which initiates many signal transduction pathways. In particular, TLR4 receptor binding can induce tyrosine and serine/threonine kinase cascades that activate transcription factors, thereby leading to the expression of inflammatory cytokines and the NOS2 gene. The inducible form of NOS catalyzes the synthesis of NO from L-arginine in response to infection. [doi: 10.1152/ajpcell.00010.2004.].

  1. Suggestions: Line 231 change “provoked” into “developed”. Line 310 change “provocation” into “status”. Line 342 change “expression” into “production”. Line 418 and 642 change “prohibition” into “inhibition”. Line 450 change “prohibit” into “inhibit”. Line 654 change “expression” into “production”. Line 655 change “dose” into “concentration”.

Responses to Comments:

Thank you very much for your critical comments. The words “provoked” (Line 352), “provocation” (Line 480), “prohibition” (Line 613, 715, 916), “prohibit” (Line 552, 646), “expression” (Line 938), “dose” (Line 939) in text were modified as suggestion. For the word “expression” (Line 512), we did not change because it was used by the authors to describe the presence of H2O2 in rat tissues via histological images [https://doi.org/10.1016/j.archoralbio.2007.10.005].

Round 2

Reviewer 1 Report

Several points of my critical remarks were considered by the authors and adequate changes were performed in the revised version of the manuscript.

However, I miss some information, which I asked to include. This missing data concerns the indication of thiocyanate level in saliva in contrast to serum level. In many papers of saliva composition, it well known that saliva thiocyanate is in low millimolar concentration, while serum levels are much lower. To indicate some selected data about SCN- concentration in saliva, is necessary to underline the significance of hypothiocyanite formation by peroxidases.

I also disagree with the answer to my third comment. Of course, chloride is present in the saliva. However, in the presence of millimolar concentrations of SCN-, it is unlikely that myeloperoxidase will form HOCl. It well known in literature that under these conditions, the oxidation of SCN- by the MPO/H2O2 system predominates.

Line 347: In the revised version, it is stated that LPO oxidizes bromide in the presence of H2O2. This is not true.

Author Response

Manuscript ID: Antioxidants-999518

Dear Reviewer,

Thank you very much for your critical comments on our manuscript. You pointed out a few more issues requiring further modifications, thereby improving the scientific content in our manuscript. We have revised our manuscript according to your comments (highlighted in yellow). Please refer to the list of Responses to Comments in the below pages.

We hope that the revised manuscript will be satisfactory for publication in the Antioxidants, and look forward to hearing from you about the acceptability of the paper. Thank you so much for your considerations.

With best regards,

I-Ta Lee, Ph.D.

School of Dentistry, College of Oral Medicine, Taipei Medical University, Taipei, Taiwan

250 Wuxing St. Taipei 11031, Taiwan

Tel: +886-2-27361661 ext. 5162

Fax: +886-2-27362295

E-mail addresses: itlee0128@tmu.edu.tw

  1. The missing data concerns the indication of thiocyanate level in saliva in contrast to serum level. In many papers of saliva composition, it well known that saliva thiocyanate is in low millimolar concentration, while serum levels are much lower. To indicate some selected data about SCN- concentration in saliva is necessary to underline the significance of hypothiocyanite formation by peroxidases.

Responses to Comments:

Thank you very much for your critical comments. The presence of significantly higher concentrations of SCN− in saliva, whereby underlining the importance of peroxidases was presented (Line 415-423).

  1. Of course, chloride is present in the saliva. However, in the presence of millimolar concentrations of SCN-, it is unlikely that myeloperoxidase will form HOCl. It well known in literature that under these conditions, the oxidation of SCN- by the MPO/H2O2 system predominates.

Responses to Comments:

        Thank you very much for your critical comments. We agree with the viewpoint that SCN− has a much higher specificity for MPO than Cl−; and SCN− can become a competitive substrate for MPO in secretions where its levels are much higher like saliva. The statement regarding the role of MPO in saliva was modified to underline the predominance of SCN− (Line 405-407).  

  1. Line 347: In the revised version, it is stated that LPO oxidizes bromide in the presence of H2O2. This is not true.

Responses to Comments:

        Thank you very much for your critical comment. Our statement was written based on another paper which presented that “LPO forms the LPO system together with thiocyanate ions or iodides or bromides and hydrogen peroxide. The mechanism of action of this system is based on oxidation of thiocyanate ions (also iodides and bromides) to hypothiocyanite ions (hypoiodides and hypobromides) with the use of hydrogen peroxide.” [doi: 10.3390/ijms20061443]. However, to focus on the role of SCN− in the LPO activity, we deleted the involvement of other halides (Line 397-398).
